# The Correlation of Apolipoprotein B with Alterations in Specific Fat Depots Content in Adults

**DOI:** 10.3390/ijms24076310

**Published:** 2023-03-27

**Authors:** Junye Chen, Kang Li, Jiang Shao, Zhichao Lai, Yuyao Feng, Bao Liu

**Affiliations:** 1Department of Vascular Surgery, Peking Union Medical College Hospital, Chinese Academy of Medical Science & Peking Union Medical College, Beijing 100730, China; pumc_jc@student.pumc.edu.cn (J.C.); likang9010@163.com (K.L.);; 2State Key Laboratory of Medical Molecular Biology, Institute of Basic Medical Sciences, Chinese Academy of Medical Sciences, Department of Pathophysiology, Peking Union Medical College, Beijing 100730, China

**Keywords:** ApoB, fat depots, dual-energy X-ray absorption, cardiovascular disease

## Abstract

Body mass index (BMI) and blood biomarkers are not enough to predict cardiovascular disease risk. Apolipoprotein B was identified to be associated with cardiovascular disease (CVD) progression. The Dual-energy X-ray Absorption (DXA) results could be considered as a predictor for cardiovascular disease in a more refined way based on fat distribution. The prediction of CVD risk by simple indicators still cannot meet clinical needs. The association of ApoB with specific fat depot features remains to be explored to better co-predict cardiovascular disease risk. An amount of 5997 adults from National Health and Nutrition Examination Survey (NHANES) were enrolled. Their demographic information, baseline clinical condition, blood examination, and DXA physical examination data were collected. Multivariate regression was used to assess the correlation between ApoB and site-specific fat characteristics through different adjusted models. Smooth curve fittings and threshold analysis were used to discover the turning points with 95% confidence intervals. ApoB is positively correlated with arms percent fat, legs percent fat, trunk percent fat, android percent fat, gynoid percent fat, arm circumference and waist circumference after adjustment with covariates for age, gender, race, hypertension, diabetes, hyperlipidemia, coronary heart disease, smoking status and vigorous work activity. The smooth curve fitting and threshold analysis also showed that depot-specific fat had lower turning points of ApoB in both males and females within the normal reference range of ApoB. Meanwhile, females have a lower increase in ApoB per 1% total percent fat and android percent fat than males before the turning points, while females have a higher growth of ApoB per 1% gynoid percent fat than males. The combined specific fat-depot DXA and ApoB analysis could indicate the risk of CVD in advance of lipid biomarkers or DXA alone.

## 1. Introduction

From 1999 to 2018, Americans with adiposity assessed as poor (body mass index, BMI ≥30, waist circumference ≥88 cm or ≥102 cm for females and males, respectively) have gradually increased from 40%~ to 60%~, according to National Health and Nutrition Examination Survey (NHANES) data [1]. However, poor blood lipids (total cholesterol: high-density lipoprotein >5:1) decreased from 1999 to 2018, while the prevalence of cardiovascular disease (CVD, such as myocardial infarction, coronary heart disease, heart failure or stroke), angina and absence of CVD stayed stable, suggesting that BMI alone could not well indicate the risk of CVD [1]. Although weight gain is associated with a higher likelihood of finding changes in CVD risk factors, not every individual with overweight or obesity shows changes in CVD risk factors due to excessive body fat. People with similar body weight or BMI may vary from different levels of complications and health risks [2]. High levels of upper body (or abdominal) fat are assessed by the waist-to-hip ratio, indicating an increased risk of impaired glucose tolerance, which is associated with insulin resistance, hypertension, hypertriglyceridemia and atrial fibrillation [3,4]. Trunk fat weight is a predictor of adverse cardiovascular metabolic characteristics, while an increase in leg fat weight is associated with a lower risk of metabolic disorders [4,5]. Postmenopausal women are more likely to have metabolic changes, partly due to the transfer of subcutaneous fat to visceral fat [6]. Chen et al. found that increased body fat and reduced leg fat were associated with an increased risk of CVD in postmenopausal women with normal BMI [7]. This evidence highlights the potential importance of fat distribution in the development of cardiac metabolic diseases.

Blood lipids are one of the most common clinical CVD screening indicators. Patients with low-density lipoprotein cholesterol (LDL-C) ≥4.9 mmol/L need to strengthen lipid reduction, anti-platelet therapies and genetic evaluation [8]. Apolipoprotein B (ApoB) was considered to be the main lipoprotein component that leads to the risk of peripheral artery disease and coronary artery disease [9]. It can be detected in LDL, very low-density lipoprotein (VLDL), chylomicron and lipoprotein(a)s in plasma, which is associated with the aggravation of atherosclerosis. It has been shown that the total number of ApoB particles has a stronger correlation with CVD risk than LDL-C [10,11], and the benefits of lipid-lowering therapy have been shown to be proportional to the decrease in ApoB [12,13,14]. Furthermore, arterial wall deposition of ApoB runs through the whole process of atherosclerosis [15]. Thereby, ApoB summarizes the results of other blood lipid indicators and may improve the clinical evaluation and treatment of atherosclerotic dyslipidemia [16].

The development of imaging technology has made it possible to accurately assess body composition and body fat distribution phenotype. Currently, cross-sectional images of the human body obtained by CT or MRI can accurately measure global and local body composition and explore the relationship between ectopic fat depots and cardiac metabolic risk factors [17]. Dual-energy X-ray absorption (DXA) can provide estimates of total fat and regional fat distribution. Currently, measuring visceral fat by DXA has been validated with the measurement methods of CT and MRI [2]. Furthermore, upper body fat depots (including android and visceral fat) and lower body fat depots (including gynoid and leg fat) showed opposite associations with CVD risks [4].

At present, there are still a few large sample studies of randomized control trials for ApoB. Our study analyzed the correlation between ApoB and fat proportion in different depots of certain American people, which can be used to guide clinical monitoring of different fat depots and CVD risk.

## 2. Results

A total of 5997 participants over the age of 18 were enrolled in this study, including 2933 females and 3064 males. Weighted Gender-specific baseline demographic characteristics are presented in Table 1. Compared with females, males had higher systolic blood pressure, diastolic blood pressure, arm circumference, waist circumference, ApoB level, triglyceride level, coronary heart disease status percentage, vigorous work activity percentage and smoking status percentage, and lower HDL-C level and high education level.

### 2.1. Associations between ApoB and Fat Distribution

DXA was used to measure the percent of fat in different fat regions. A positive correlation between ApoB and fat distribution was observed (Table 2). When no covariate was adjusted, arms percent fat, trunk percent fat, android percent fat, arm circumference and waist circumference were positively associated with ApoB level. Among them, arm circumference (β = 1.06, 95%CI: 0.94, 1.19; *p* ≤ 0.0001) and trunk percent fat (β = 0.69, 95%CI: 0.62, 0.77; *p* ≤ 0.0001) had the biggest increase. Legs percent fat (β = 0.00, 95%CI: −0.06, 0.07; *p* = 0.9242) and gynoid percent fat (β = 0.00, 95%CI: −0.08, 0.07; *p* = 0.9381) had no correlation with ApoB level. After adjustment of age, gender, race, hypertension, diabetes, hyperlipidemia, coronary heart disease, smoking status and vigorous work activity, legs percent fat (β = 0.33, 95%CI: 0.22, 0.44; *p* ≤ 0.0001) and gynoid percent fat (β = 0.41, 95%CI: 0.28, 0.53; *p* ≤ 0.0001) both had a positive association with blood ApoB level. Trunk percent fat (β = 0.82, 95%CI: 0.73, 0.92; *p* ≤ 0.0001), total percent fat (β = 0.77, 95%CI: 0.65, 0.88; *p* ≤ 0.0001) and arm circumference (β = 0.77, 95%CI: 0.63, 0.92; *p* ≤ 0.0001) became the largest increase indicators of ApoB level.

### 2.2. The Threshold Analyses between ApoB and Fat Distributions 

The associations between ApoB and body fat distribution (total percent fat, trunk percent fat, arms percent fat, legs percent fat, android percent fat and gynoid percent fat, arm circumference and waist circumference) were further confirmed by adjusted models and smooth curve fittings (Figure 1 and Figure 2). Although male and female ApoB intersect in the high total percent fat area and the differences tend to decrease, results from arm percent fat and arm circumference show that ApoB is still not intersecting between females and males at the level of severe obesity, which is related to fat distribution characteristics of women. For other results from trunk percent fat, leg percent fat, waist circumference, android percent fat and gynoid percent fat, although there are intersections in the case of high proportion of these parameters, the trend has gone in the opposite direction. This means that gender still plays an important role in severe obesity and needs to be analyzed according to specific locations. It also shows the importance of the joint application of DXA and ApoB.

Non-linear relationships of ApoB with fat distribution features were detected. When stratified by gender, the turning points of the non-linear relationships were further calculated. Male: Arms percent fat (%): 24.8%, Legs percent fat (%): 26.4%, Trunk percent fat (%): 29.9%, Total percent fat (%): 27.8%, Android percent fat (%): 30.1%, Gynoid percent fat (%): 27.8%, Arm circumference (cm): 31.9 cm, Waist circumference (cm): 99.6 cm; Females: Arms percent fat (%): 49.1%, Legs percent fat (%): 41.05%, Trunk percent fat (%): 45.8%, Total percent fat (%): 39.5%, Android percent fat (%): 45.4%, Gynoid percent fat (%): 39.9%, Arm circumference (cm): 33.9 cm, Waist circumference (cm): 102.6 cm (Table 3). 

Detailed speaking, women have a lower increase in ApoB (Female: 1.31 mg/dL, 95%CI: 1.06~1.56; male: 1.77 mg/dL, 95%CI: 1.44~2.09) per 1% total percent fat (<39.5% for female; <24.8% for male) than men before the turning point. Women have a lower increase in ApoB (Female: 0.80 mg/dL, 95%CI: 0.67~0.93; male: 1.36 mg/dL, 95%CI: 1.10~1.61) per 1% android percent fat (<45.4% for female; <30.1% for male) than men before the turning point, while women have a higher growth of ApoB (Female: 1.32 mg/dL, 95%CI: 0.93~1.70; male: 1.20 mg/dL, 95%CI: 0.85~1.55) per 1% gynoid percent fat (<39.9% for female; <27.8% for male) than men. At the same time, the association between ApoB and waist circumference and gynoid percent fat were both over-fitted in men, suggesting that the joint analyses of ApoB with waist circumference and gynoid percent fat are more suitable for women to predict the risk of cardiovascular disease.

## 3. Discussion

The results of this study showed ApoB was positively correlated with distinct fat distribution in a nationally representative sample of U.S. adults. Specifically, after being adjusted by various models including age, gender, race, hypertension, diabetes, hyperlipidemia, coronary heart disease, smoking status and vigorous work activity, ApoB was positively associated with DXA body fat distribution indicators, including arms percent fat, legs percent fat, trunk percent fat, total percent fat, android percent fat, gynoid percent fat, arm circumference and waist circumference. Meanwhile, the nonlinear relationships between ApoB and body fat distribution stratified by gender were further identified through adjusted models and spline smoothing and the turning points were calculated. Compared with the normal reference value of ApoB < 130 mg/dl, the model fit value at turning points of different fat depots of DXA is within the normal reference range. The combined monitoring of DXA and ApoB can predict cardiovascular risk more accurately and prevent cardiovascular events as early as possible.

### 3.1. Apolipoprotein B and CVD

Apolipoprotein B (ApoB) can be detected in LDL, very low-density lipoprotein (VLDL), chylomicron and lipoprotein(a)s in plasma and leads to the risk of peripheral artery disease and coronary artery disease by depositing in the arterial wall throughout the whole process of atherosclerosis [9,15]. It has been shown that the total number of ApoB particles has a stronger correlation with CVD risk than LDL-C [10,11]. Lim et al. [18] demonstrated that high under-curve area of plasma ApoB was independently related to the increase in intima-media thickness of the carotid artery and low ankle/brachial index (ABI). A recent analysis of Dorobanțu et al. [19] on the Romanian National Registry of Hypertension suggests that high ApoB could be considered a risk factor for CVD and be associated with classic markers of clinical or subclinical CVD, including altered lipid profiles, poor glycemic control, significant carotid plaque and elevated inflammatory status.

### 3.2. Apolipoprotein B and Depot-Specific Body Fat

Overweight and obesity increase the risk of obtaining CVD [20,21,22]. DXA could directly obtain the distribution of body fat noninvasively to evaluate the correlation between specific fat depots and cardiovascular disease risk [23,24]. Upper-body and lower-body fat accumulation indicated the opposite correlation with cardiovascular disease [3]. Abdominal fat reserves are characterized by rapid intake of dietary fat, as well as high-fat turnover, which is easily stimulated by adrenergic receptor activation [25], while lower-body fat plays a protective role with a reduced fat turnover rate and it is able to accommodate the fat redistribution by retaining the ability to recruit extra fat cells as a result of weight gain to form pear-shaped bodies, as well as metabolically-healthy obesity [26]. The functional differences between upper- and lower-body fat tissues are controlled by certain depot-specific transcription factors, such as HOXA3, HOXA5, HOXA6, IRX2 and TBX5 in abdominal subcutaneous adipose tissue [27,28] and HOTAIR, SHOX2 and HOXC11 in gluteofemoral adipose tissue [27,29].

At present, there are few studies focused on the correlation between ApoB and fat accumulation in different locations. Fat is the main source of energy in the human body. The absorption and redistribution of fat require the installation of lipoproteins containing ApoB in the intestine and liver [30]. By contrast, the rodent liver produces a mixture of ApoB100 and ApoB48, while the human liver produces only ApoB100 [31]. In intestinal-derived lipoproteins, ApoB48 far exceeds ApoB100. The primary apolipoprotein secreted by the liver is very low-density lipoprotein (VLDL), which plays a central role in the peripheral deposition of hepatic or intestinal-derived fatty acids [32]. Therefore, ApoB can be considered a potential therapeutic target for obesity. Based on the ApoB100 mimic epitope, the polypeptide pB1-based vaccine-like formulations (BVF) can reduce the weight gain of mice caused by a 3-month high-fat diet by 44% and 65%, preventing mesenteric fat accumulation and liver steatosis [33]. 

In multivariate regression, after correcting baseline TG and REE (resting energy expenditure base), high abdominal fat area and low HDL concentration were independently correlated with the area under the curve of ApoB, while the area under the curve of high ApoB was independently related to the increase in intima-media thickness of the carotid artery and low ankle/brachial index (ABI) [18]. Furthermore, lipolysis rates were negatively correlated with plasma ApoB levels in human subcutaneous adipocytes, but only LDL was found to inhibit lipolysis in a concentration-dependent manner, while VLDL had no effect on lipolysis [34]. Norreen-Thorsen et al. analyzed the transcriptome expression of *APOB* in 527 cases of visceral fat and 646 cases of subcutaneous fat [35]. It was found that the mean transcripts per million (TPM, mean expression across all samples) of visceral fat and subcutaneous fat were 1.54 and 1.20, respectively. It is suggested that ApoB is negatively correlated with lipolysis and positively correlated with fat accumulation in visceral fat and subcutaneous fat, but has a more significant correlation with abdominal fat.

### 3.3. Apolipoprotein B and Gender

The total body fat of women is higher than men, and women most often accumulate adipose tissue subcutaneously, while men and postmenopausal women tend to accumulate adipose tissue in the central viscera [36,37]. It has been found that the cellular mechanism regulating the proliferation and function of male and female adipocytes is driven by cellular autonomous properties related to sex chromosome complement and structure [38]. Most of a female’s fat (80%~90%) is stored in subcutaneous fat depots, and the fat distribution is more like a pear shape due to the expansion of the lower body, buttocks and femur fat depots, which is consistent with the results of this study that the turning points of arms percent fat and legs percent fat of females are higher than that of males. The enlargement of subcutaneous fat in the lower body of females is associated with protection from glucose-insulin homeostasis and hypertriglyceridemia [39]. Compared with men, women’s lower body adipose tissue absorbs triglycerides-fatty acids from food through lipoprotein lipase and free fatty acids directly from circulation more effectively [40]. Women’s lower body fat is considered a “safe repository” that can be expanded by recruiting adipose precursor cells and preadipocytes to prevent ectopic fat deposits in muscles and liver and excessive visceral fat [37]. Estrogen prevents increased body fat and obesity by suppressing appetite and increasing energy consumption. Estradiol inhibits food intake by increasing the potency of other anorexia signals, such as ApoB [41], leptin, and by reducing appetite signals [42], which may also be one of the mechanisms by which lower body fat plays a protective role. Norreen-Thorsen et al. [35] analyzed the transcriptional expression of visceral fat samples in 165 cases and subcutaneous fat samples in 212 cases of females and the transcriptional expression of visceral fat samples in 362 cases and subcutaneous fat samples in 434 cases of males. It was found that the *APOB* mean TPM of visceral fat was 2.28 in males and 1.20 in females. The *APOB* mean TPM of subcutaneous fat was 1.55 in men and 1.03 in women. These studies are also consistent with the findings of our study that women have a lower increase in ApoB per 1% total percent fat and android percent fat than men before the turning point, while women have a higher growth of ApoB per 1% gynoid percent fat than men. 

According to our findings, women with a total percent fat less than 39.5% and a plasma ApoB level greater than 95.66 mg/dL, and men with a total percent fat less than 27.8% and a plasma ApoB level greater than 102.40 mg/dL, should be considered to have preclinical atherosclerosis and receive appropriate lifestyle education. The android and gynoid methods can be used to identify preclinical atherosclerosis. Men with an android percent fat less than 30.1% and a plasma ApoB level greater than 100.57 mg/dL, and women with a gynoid percent fat less than 39.9% and a plasma ApoB level greater than 93.65 mg/dL, should receive appropriate guidance for preclinical atherosclerosis. If DXA is not available, a combined approach of waist and arm circumference with ApoB can be considered. Men with an arm circumference less than 31.9 cm and a plasma ApoB level greater than 98.04 mg/dL, and women with a waist circumference less than 102.6 cm and a plasma ApoB level greater than 99.96 mg/dL, should be considered to have preclinical atherosclerosis and receive corresponding lifestyle education.

### 3.4. Advantages and Limitations

The main advantage of this study is that it obtained a large number of nationally representative groups of American adults, who have complete data of NHANES on different fat depots of DXA data, lipid biomarkers and clinical baseline data. Image quantitative analysis of different fat depots combined with circulating ApoB data will provide a more accurate cardiovascular disease risk prediction program. 

This study has some limitations. In the American adult population in this study, the number of people diagnosed with cardiovascular disease is relatively small, so it is not possible to analyze the correlation between patients and healthy controls, but can only use the known correlation between elevated ApoB and cardiovascular disease and the correlation between obesity and cardiovascular disease to build a joint prediction model of specific fat depot content and cardiovascular disease risks. A large sample of correlation analysis between cardiovascular patients and healthy controls is still needed to popularize this model. This study only briefly discussed the mechanism of the correlation between ApoB and fat accumulation in specific fat depots, but there is still a lack of further research. In this study, the correlation between ApoB and gender on fat accumulation in different fat depots is also only briefly discussed, and further research remains needed. 

## 4. Materials and Methods

### 4.1. Study Population

The clinical data of 116,863 people in 11 cycles of the NHANES database from 1999 to 2020 were obtained. The study was approved by the National Center for Health Statistics Ethics Review Board (https://www.cdc.gov/nchs/nhanes/irba98.htm, accessed on 24 August 2022). Written informed consent was obtained from all participants. NHANES is a project to assess the health and nutritional status of adults and children in the United States through interviews and physical examinations. The interviews included demographic, socioeconomic and health-related issues, while the examination section included physiological and medical measurements conducted by trained medical staff using standardized instruments. Among the 116,863 people who participated in the NHANES survey, a total of 68,896 adults (≥18 years old) were included for further analysis. Patients without total percent fat data (*n* = 38,320) and those without ApoB data (*n* = 32,323) were excluded and the remaining 5997 entered the final analysis.

### 4.2. Variables

Gender, age, race, education, income, marital status, blood pressure, BMI, arm circumference, waist circumference, total cholesterol, ApoB, HDL-C, TG, LDL-C, hypertension (doctor ever said you had high blood pressure), hyperlipidemia (doctor ever said you had high cholesterol level), diabetes (doctor ever said you had diabetes), coronary heart disease (doctor ever said you had coronary heart disease), vigorous work activity, smoked at least 100 cigarettes in life and current smoking status (do you now smoke cigarettes) were collected as baseline data. Arms percent fat (%), legs percent fat (%), trunk percent fat (%), total percent fat (%), android percent fat (%), gynoid percent fat (%), arm circumference (cm) and waist circumference (cm) were included as exposure variables. Detailed laboratory protocols could be accessed by NHANES website (http://www.cdc.gov/nchs/nhanes/, accessed on 14 August 2022).

Body fat composition was measured by DXA (Hologic Horizon QDR 4500A fan-beam bone densitometer) following a standard protocol [43]. The Hologic APEX software was used to define the android and gynoid regions. The android area (A) is located at the area around the waist between the midpoint of the lumbar spine and both tops of the pelvis; the gynoid area (G) is located between the femoral head and middle thigh at both sides (Figure 3). Analysis of the whole body scans could provide fat depot-specific measurements for the arms, legs, trunk, android, gynoid and total mass of fat and percent fat (%).

Blood pressure: After 5 min of rest, the maximum inflation level and three consecutive blood pressure readings were determined. All blood pressure measurements (systolic and diastolic blood pressure) were performed at the Mobile Examination Center (MEC).

Body measures: The body measurement data were collected by trained technicians, accompanied by a recorder during each physical measurement. As a result, body weight data for amputees were set to missing. All MEC body measurement rooms were identical in terms of equipment layout and type. Scheduled equipment calibration was carried out and verified by supervisors.

Vigorous work activity: The intensity of the subject’s work resulted in a substantial increase in breathing or heart rate.

Laboratory data: Blood specimens are processed and stored at −30 °C and eventually shipped to the University of Minnesota for testing. Serum samples were measured using the Roche Hitachi 717, 912 and Roche Modular P chemistry analyzer. Detailed specimen collection and processing instructions are discussed in the NHANES Laboratory/Medical Technologists Procedures Manual (LPM).

### 4.3. Statistical Analysis

The 2-year sample weights (WTINT2YR or WTMEC2YR) were used for all NHANES analyses as one circle contains 2 years of data. All analyses were performed with EmpowerStats (http://www.empowerstats.com, accessed on 14 August 2022) by calling package R (http://www.Rproject.org, accessed on 14 August 2022). The significance of statistics was considered when *p* value < 0.05. Multivariable linear regression models were conducted to assess the associations between ApoB and total percent fat, trunk percent fat, arms percent fat, legs percent fat, android percent fat and gynoid percent fat, arm circumference and waist circumference. Three models were built for multivariable linear regression: unadjusted model 1, adjusted model 2 (adjusted for age, gender and race) and adjusted model 3 (adjusted for age, gender, race, hypertension, diabetes, hyperlipidemia, coronary heart disease, smoking status and vigorous work activity). The spline smoothing plots and adjusted models were used to address the piecewise associations between ApoB and specific fat depot features stratified by gender. Log-likelihood ratio test and two steps recursive method were used to determine the turning point (K), which gives the maximum likelihood. The confidence interval of the threshold was determined by the bootstrap resampling method.

## 5. Conclusions

Overall, this study found that combined analysis of depot-specific DXA indicators such as arms percent fat, legs percent fat, trunk percent fat, total percent fat, android percent fat, gynoid percent fat and ApoB would indicate the risk of cardiovascular disease earlier than lipid biomarkers alone.

## Figures and Tables

**Figure 1 ijms-24-06310-f001:**
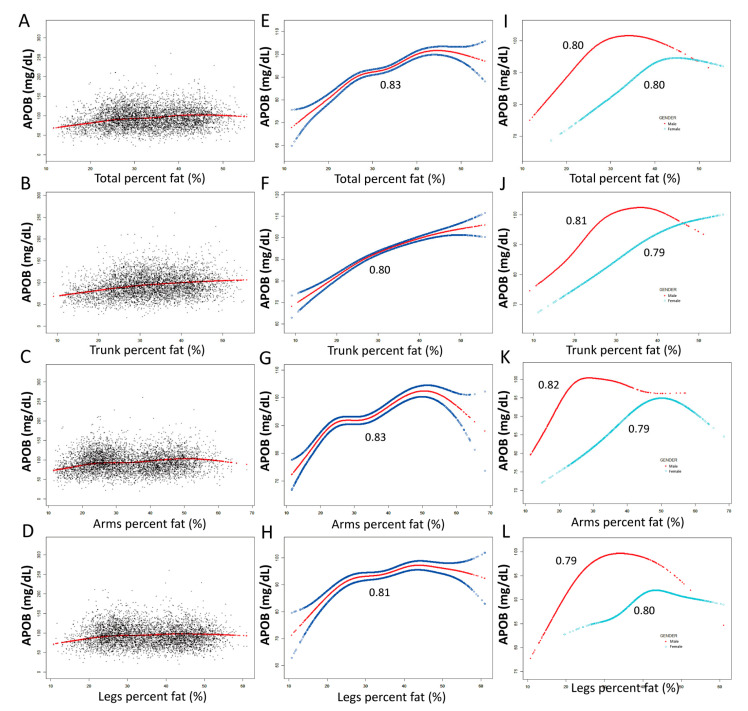
The correlation between ApoB and total percent fat, trunk percent fat, arms percent fat and legs percent fat. (**A**–**D**), black dots indicate the distribution of individual values. (**E**–**H**), red line shows the smooth curve fitting of the variables, and the area between the two blue lines represents 95% of the confidence interval (CI). (**I**–**L**), smooth curve fitting of variables stratified by gender (red line for male, blue line for female). Age, gender, race, hypertension, diabetes, hyperlipidemia, coronary heart disease, smoking status and vigorous work activity were adjusted. (**I**–**L**) were not adjusted by gender.

**Figure 2 ijms-24-06310-f002:**
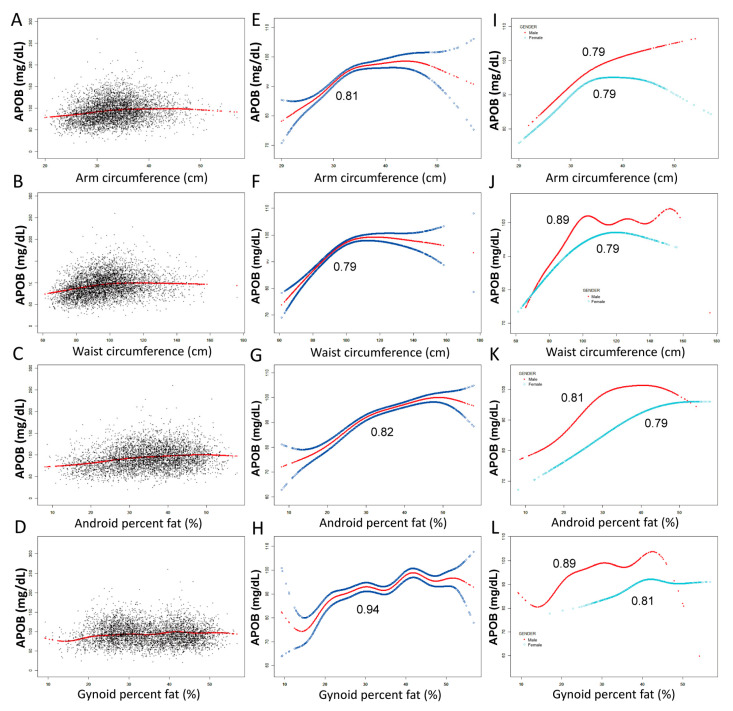
The correlation between ApoB and arm circumference, waist circumference, android percent fat and gynoid percent fat. (**A**–**D**), black dots indicate the distribution of individual values. (**E**–**H**), red line shows the smooth curve fitting of the variables, and the area between the two blue lines represents 95% of the confidence interval (CI). (**I**–**L**), smooth curve fitting of variables stratified by gender (red line for male, blue line for female). Age, gender, race, hypertension, diabetes, hyperlipidemia, coronary heart disease, smoking status and vigorous work activity were adjusted. (**I**–**L**) were not adjusted by gender.

**Figure 3 ijms-24-06310-f003:**
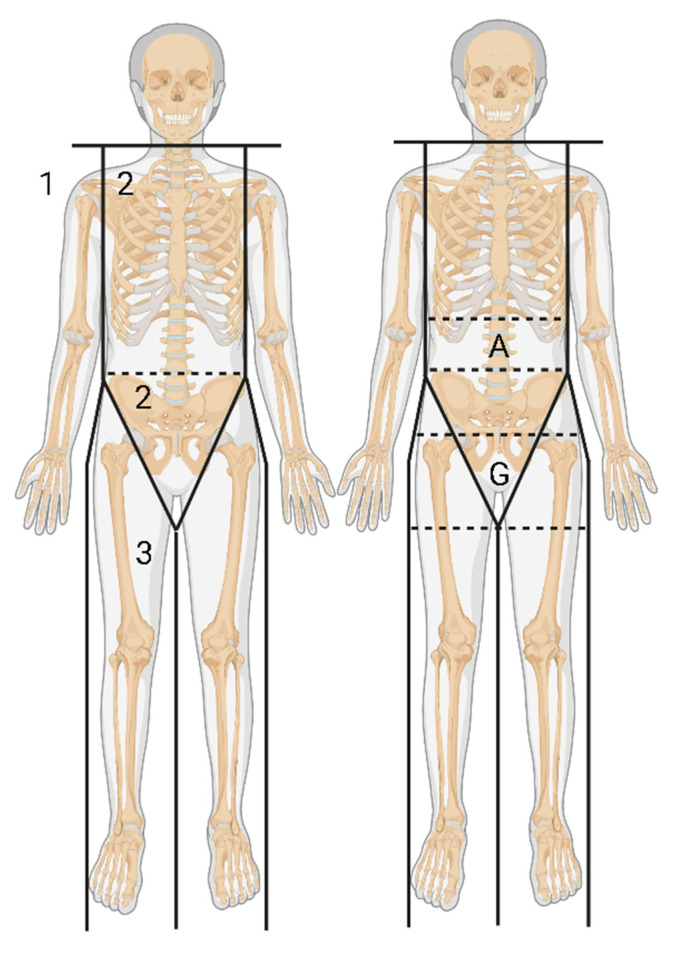
Fat distribution regions measured by dual-energy x-ray absorptiometry. 1. arm fat region; 2. trunk fat region; 3. leg fat region; A. android region; G. gynoid region. Created with Biorender.com.

**Table 1 ijms-24-06310-t001:** Clinical demographic characteristics. Mean ± SD for continuous variables; Percentage (%) for categorical variables; BMI, body mass index; ApoB, apolipoprotein B; HDL, high-density lipoprotein; LDL, low-density lipoprotein; AA, associates.

	Males (3064)	Females (2933)
Age (years)	39.45 ± 12.93	40.43 ± 12.77
Ratio of family income to poverty	2.98 ± 1.64	2.94 ± 1.67
Systolic blood pressure (mmHg)	121.40 ± 14.28	116.09 ± 14.99
Diastolic blood pressure (mmHg)	71.42 ± 12.10	69.21 ± 10.15
BMI (kg/m^2^)	28.43 ± 6.04	28.85 ± 7.45
Arm circumference (cm)	34.36 ± 4.44	32.12 ± 5.46
Waist circumference (cm)	99.51 ± 15.98	95.02 ± 16.94
Total cholesterol (mmol/L)	4.94 ± 1.05	5.01 ± 1.06
ApoB (mg/dL)	96.18 ± 26.84	91.07 ± 25.64
HDL-C (mmol/L)	1.25 ± 0.35	1.53 ± 0.41
Triglyceride (mmol/L)	1.61 ± 1.34	1.25 ± 1.19
LDL-C (mmol/L)	2.98 ± 0.91	2.92 ± 0.89
Race (%)		
Mexican American	10.79	8.78
Hispanic	6.43	6.04
Non-Hispanic White	65.23	64.48
Non-Hispanic Black	9.99	12.44
Others	7.58	8.26
Education level (%)		
Less than high school	16.61	13.42
High school grad or equivalent	24.79	19.47
AA degree/college graduate or above	58.60	67.11
Marital status (%)		
Married or with a partner	63.98	64.19
Unmarried and other	36.02	35.81
Hypertension (%)		
Yes	24.99	23.35
No	75.01	76.65
Hyperlipidemia (%)		
Yes	30.63	28.03
No	69.37	71.97
Diabetes (%)		
Yes	5.71	6.57
No	94.29	93.43
Coronary heart disease (%)		
Yes	2.17	0.74
No	97.83	99.26
Vigorous work activity (%)		
Yes	36.52	21.83
No	63.48	78.17
Smoked at least 100 cigarettes in life		
Yes	51.47	37.50
No	48.53	62.50
Current smoking status (%)		
Yes	54.60	55.60
No	45.40	44.40

**Table 2 ijms-24-06310-t002:** Correlation between ApoB and specific fat depot features. Model 1, no covariate was adjusted; Model 2, age, gender, and race were adjusted; Model 3, age, gender, race, hypertension, diabetes, hyperlipidemia, coronary heart disease, smoking status and vigorous work activity were adjusted.

Exposures	ApoB (mg/dL)		
	Model 1 β (95%CI) *p*-Value	Model 2 β (95%CI) *p*-Value	Model 3 β (95%CI) *p*-Value
Arms percent fat (%)	0.19 (0.13, 0.24) <0.0001	0.56 (0.48, 0.64) <0.0001	0.54 (0.44, 0.63) <0.0001
Legs percent fat (%)	0.00 (−0.06, 0.07) 0.9242	0.36 (0.26, 0.46) <0.0001	0.33 (0.22, 0.44) <0.0001
Trunk percent fat (%)	0.69 (0.62, 0.77) <0.0001	0.83 (0.75, 0.92) <0.0001	0.82 (0.73, 0.92) <0.0001
Total percent fat (%)	0.40 (0.32, 0.48) <0.0001	0.80 (0.69, 0.90) <0.0001	0.77 (0.65, 0.88) <0.0001
Android percent fat (%)	0.63 (0.56, 0.70) <0.0001	0.64 (0.57, 0.72) <0.0001	0.66 (0.57, 0.74) <0.0001
Gynoid percent fat (%)	−0.00 (−0.08, 0.07) 0.9381	0.38 (0.27, 0.49) <0.0001	0.41 (0.28, 0.53) <0.0001
Arm circumference (cm)	1.06 (0.94, 1.19) <0.0001	0.85 (0.72, 0.98) <0.0001	0.77 (0.63, 0.92) <0.0001
Waist circumference (cm)	0.42 (0.38, 0.46) <0.0001	0.30 (0.26, 0.34) <0.0001	0.28 (0.24, 0.33) <0.0001

**Table 3 ijms-24-06310-t003:** Threshold analysis of ApoB and specific fat depot features. Age, race, hypertension, diabetes, hyperlipidemia, coronary heart diseases, vigorous work activity and smoking status were adjusted.

Exposures	Turning Point	<Turning Point β (95%CI) *p*-Value	>Turning Point β (95%CI) *p*-Value	Model Fit Value at K	Log Likelihood Ratio
	ApoB (mg/dL)				
Males					
Arms percent fat (%)	24.8	1.81 (1.45, 2.16) < 0.0001	−0.08 (−0.31, 0.16) 0.5329	102.17 (100.57, 103.78)	<0.001
Legs percent fat (%)	26.4	1.36 (0.96, 1.76) < 0.0001	0.05 (−0.20, 0.31) 0.6811	99.73 (98.07, 101.40)	<0.001
Trunk percent fat (%)	29.9	1.52 (1.28, 1.75) < 0.0001	−0.15 (−0.47, 0.18) 0.3677	104.21 (102.55, 105.86)	<0.001
Total percent fat (%)	27.8	1.77 (1.44, 2.09) < 0.0001	−0.01 (−0.32, 0.30) 0.9414	102.40 (100.77, 104.04)	<0.001
Android percent fat (%)	30.1	1.36 (1.10, 1.61) < 0.0001	0.14 (−0.10, 0.37) 0.2554	100.57 (98.81, 102.34)	<0.001
Gynoid percent fat (%)	27.8	1.20 (0.85, 1.55) < 0.0001	−0.07 (−0.39, 0.25) 0.6520	100.26 (98.60, 101.91)	<0.001
Arm circumference (cm)	31.9	2.46 (1.71, 3.20) < 0.0001	0.49 (0.18, 0.80) 0.0018	98.04 (96.34, 99.74)	<0.001
Waist circumference (cm)	99.6	0.78 (0.63, 0.93) < 0.0001	0.04 (−0.06, 0.15) 0.4293	102.95 (101.35, 104.55)	<0.001
Females					
Arms percent fat (%)	49.1	0.80 (0.65, 0.95) < 0.0001	−0.95 (−1.37, −0.52) < 0.0001	100.11 (98.43, 101.78)	<0.001
Legs percent fat (%)	41.05	0.95 (0.62, 1.27) < 0.0001	−0.40 (−0.66, −0.14) 0.0027	93.38 (91.60, 95.16)	<0.001
Trunk percent fat (%)	45.8	0.93 (0.79, 1.06) < 0.0001	−0.95 (−1.80, −0.11) 0.0274	103.05 (101.30, 104.79)	<0.001
Total percent fat (%)	39.5	1.31 (1.06, 1.56) < 0.0001	−0.19 (−0.52, 0.14) 0.2515	95.66 (93.97, 97.34)	<0.001
Android percent fat (%)	45.4	0.80 (0.67, 0.93) < 0.0001	−0.67 (−1.22, −0.11) 0.0190	99.78 (98.10, 101.45)	<0.001
Gynoid percent fat (%)	39.9	1.32 (0.93, 1.70) < 0.0001	−0.42 (−0.72, −0.12) 0.0065	93.65 (91.84, 95.46)	<0.001
Arm circumference (cm)	33.9	1.59 (1.26, 1.92) < 0.0001	−0.40 (−0.75, −0.04) 0.0295	99.03 (97.26, 100.81)	<0.001
Waist circumference (cm)	102.6	0.53 (0.43, 0.63) < 0.0001	−0.10 (−0.22, 0.03) 0.1259	99.96 (98.12, 101.80)	<0.001

## Data Availability

The data that support the findings of this study are available from the corresponding author upon reasonable request.

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
