# Peer review of "The Correlation of Apolipoprotein B with Alterations in Specific Fat Depots Content in Adults"

_ijms, 2023, doi:10.3390/ijms24076310_

Round 1

Reviewer 1 Report

The present paper is an interesting one, regarding the predictive role of apoB and depot-specific DXA indicators in predicting cardiovascular risk. However, in order to consider it for publishing, the following should be taken into account:

1.      Re-check the abbreviations. Line 58-59 - apolipoprotein B requires the use of the abbreviated form, already defined in the text above.

2.      Never use abbreviated forms at the beginning of a sentence.

3.      Reword lines 73-78. Define clear and short objectives rather than specifying what analyzes you have done. Do not enter data about the study population in defining the objectives.

4.      Lines 102-110 require the addition of the bibliographic source(s) or reformulate the entire paragraph according to the following model: lines 109-110 “Analysis of whole-body scans was performed in order to obtain fat depot-specific measurements for the arms, legs, trunk, android, gynoid and total mass of fat and percent fat.”

5.      I suggest dividing the discussion section into subsections for easier reading.

6.      I suggest discussing interesting results recently published in IJMS: DorobanÈ›u, Maria et al. “The Association between Apolipoprotein B, Cardiovascular Risk Factors and Subclinical Atherosclerosis-Findings from the SEPHAR National Registry on Hypertension in Romania.” International journal of molecular sciences vol. 24,3 2813. 1 Feb. 2023, doi:10.3390/ijms24032813.

7.      The Limitations and Conclusions sections, defined as such, are missing from the paper, although the information is found in the text.

Author Response

Q1: Re-check the abbreviations. Line 58-59 - apolipoprotein B requires the use of the abbreviated form, already defined in the text above.

A1: Thank you for your suggestion, the corresponding area has been modified, please check the updated manuscript.

Q2: Never use abbreviated forms at the beginning of a sentence.

A2: Thank you for your suggestion. We have rewritten the corresponding area, please check the updated manuscript.

Q3: Reword lines 73-78. Define clear and short objectives rather than specifying what analyzes you have done. Do not enter data about the study population in defining the objectives.

A3: You are right. We have rewritten this part, please check the updated manuscript.

Q4: Lines 102-110 require the addition of the bibliographic source(s) or reformulate the entire paragraph according to the following model: lines 109-110 “Analysis of whole-body scans was performed in order to obtain fat depot-specific measurements for the arms, legs, trunk, android, gynoid and total mass of fat and percent fat.”

A4: Thank you for your suggestion. We have added corresponding citation and rewritten this part, please check the updated manuscript.

Q5: I suggest dividing the discussion section into subsections for easier reading.

A5: Thank you for your suggestion. We have divided the discussion into subsections, please check the updated manuscript.

Q6: I suggest discussing interesting results recently published in IJMS: DorobanÈ›u, Maria et al. “The Association between Apolipoprotein B, Cardiovascular Risk Factors and Subclinical Atherosclerosis-Findings from the SEPHAR National Registry on Hypertension in Romania.” International journal of molecular sciences vol. 24,3 2813. 1 Feb. 2023, doi:10.3390/ijms24032813.

A6: Thank you for your advice. This article has a lot of exploration on ApoB and CVD risk. ApoB was found to related to lipid profiles, poor glycemic control, significant carotid plaque and elevated inflammatory status and is thereby considered to be an important risk factor for CVD in clinical or subclinical studies. We have discussed this paper in the discussion section, please check the updated manuscript.

Q7: The Limitations and Conclusions sections, defined as such, are missing from the paper, although the information is found in the text.

A7: You are right. We have added the corresponding sections of limitations and conclusions, please check the updated manuscript.

Reviewer 2 Report

This study analyzes the correlation between ApoB and fat proportion to better co-predict cardiovascular disease risk with individuals from NHANES study.

My points:

1.    The question research needs to be clear in the abstract. What is the function of ApoB to be considered as a predictor of CVD?

2.    Line 12 says “The Dual-energy X-ray Absorption (DXA) is expected to be a risk predictor for cardiovascular disease” but I think the DXA fat distribution results could be considered a predictor of CVD. Please rewrite.

3.    The abstract has no results regarding gender, menopausal status, BMI, waist circumference, waist-to-hip ratio, and blood lipid levels. I think this information is essential to better understand the sample characteristics.

4.     Line 39, please change overweight or obese individuals to people with overweight or obesity. Medical literature no longer uses diseases as adjectives for people. Please correct all over the manuscript.

5.    In line 140, when adjusting the models, did you consider menopausal status as a confounder variable?

6.    Considering the associations between ApoB and body fat distribution, I can see in Figure 2 the total percent of fat, that the gender lines meet at BMI 48,50. How can we interpret this result? Does gender not influence anymore in people with severe obesity?

7.    I think the main result is described in lines 210-252, but this description is already seen in table 3. I think you should resume these lines and highlight the top results for men and women.

8.    Finally, in the discussion section I recommend you to better describe the clinical implications for your results. How these findings can be useful during medical appointments? Is there an ApoB cutoff point considering the gender and fat percentual? How can your finding be used in translational research, considering transforming the results into new treatments and approaches to medical care that improve the population's health?

Author Response

Q1: The question research needs to be clear in the abstract. What is the function of ApoB to be considered as a predictor of CVD?

A1: Thank you for your suggestion. We have rewritten the corresponding area, please check the updated manuscript.

Q2. Line 12 says “The Dual-energy X-ray Absorption (DXA) is expected to be a risk predictor for cardiovascular disease” but I think the DXA fat distribution results could be considered a predictor of CVD. Please rewrite.

A2: Thank you for your suggestion. We have rewritten this part, please check the updated manuscript.

Q3: The abstract has no results regarding gender, menopausal status, BMI, waist circumference, waist-to-hip ratio, and blood lipid levels. I think this information is essential to better understand the sample characteristics.

A3: Thank you for your suggestion. We have added these information in the abstract, please check the updated manuscript.

Q4: Line 39, please change overweight or obese individuals to people with overweight or obesity. Medical literature no longer uses diseases as adjectives for people. Please correct all over the manuscript.

A4: Apologize for our carelessness. We have modified this issue throughout the article, please check the updated manuscript.

Q5: In line 140, when adjusting the models, did you consider menopausal status as a confounder variable?

A5: That's a very good question. We did not include menopausal status when adjusting for confounding variables. The subgroup analysis maintains at gender level. Postmenopausal women are more likely to have metabolic changes, partly due to the transfer of subcutaneous fat to visceral fat. We plan to conduct subsequent analyses specifically targeting menopausal status, fat distribution and other CVD risk factors such as lipid profile in women.

Q6: Considering the associations between ApoB and body fat distribution, I can see in Figure 2 the total percent of fat, that the gender lines meet at BMI 48,50. How can we interpret this result? Does gender not influence anymore in people with severe obesity?

A6: Gender can still have an impact on the development and health effects of severe obesity.  The distribution of body fat can differ between genders, with men more likely to have excess visceral fat and women more likely to have excess subcutaneous fat.

For example, results from arm percent fat and arm circumference show that ApoB is still not intersecting between female and male at the level of severe obesity, which is related to fat distribution characteristics of women. Other results, For example, trunk percent fat, leg percent fat, waist circumference, android percent fat and gynoid percent fat although there is intersection in the case of high proportion of these index, but the trend has gone in the opposite direction. This means that gender still plays an important role in severe obesity and needs to be analyzed according to specific location. It also shows the importance of the joint application of DXA and ApoB.

Q7: I think the main result is described in lines 210-252, but this description is already seen in table 3. I think you should resume these lines and highlight the top results for men and women.

A7: Apologize for our carelessness. We have rewritten this part in the abstract, please check the updated manuscript.

Q8: Finally, in the discussion section I recommend you to better describe the clinical implications for your results. How these findings can be useful during medical appointments? Is there an ApoB cutoff point considering the gender and fat percentual? How can your finding be used in translational research, considering transforming the results into new treatments and approaches to medical care that improve the population's health?

A8: Very well. We have revised this part according to your comments. In this study, preclinical atherosclerosis in men and women can be identified in advance based on total percent fat combined with ApoB. Since DXA can also identify android and gynoid regions, these two partitioning methods can also be used for identification of preclinical atherosclerosis status. At the same time, the corresponding criteria for the identification of arm circumference and waist circumference in conjunction with ApoB are also listed, taking into account the absence of DXA. It provides a new method to identify the preclinical status of arteriosclerosis in advance.